# An expanded reference map of the human gut microbiome reveals hundreds of previously unknown species

Sigal Leviatan [1,2,4], Saar Shoer [1,2,4], Daphna Rothschild[1,2,4], Maria Gorodetski[3] & Eran Segal [1,2✉]

The gut is the richest ecosystem of microbes in the human body and has great influence on our health. Despite many efforts, the set of microbes inhabiting this environment is not fully known, limiting our ability to identify microbial content and to research it. In this work, we combine new microbial metagenomic assembled genomes from 51,052 samples, with previously published genomes to produce a curated set of 241,118 genomes. Based on this set, we procure a new and improved human gut microbiome reference set of 3594 high quality species genomes, which successfully matches 83.65% validation samples' reads. This improved reference set contains 310 novel species, including one that exists in 19% of validation samples. Overall, this study provides a gut microbial genome reference set that can serve as a valuable resource for further research.

[1] Department of Computer Science and Applied Mathematics, Weizmann Institute of Science, Rehovot, Israel. [2] Department of Molecular Cell Biology, Weizmann Institute of Science, Rehovot, Israel. [3] DayTwo LTD, Tel Aviv, Israel. [4]These authors contributed equally: Sigal Leviatan, Saar Shoer, Daphna Rothschild. ✉email: eran.segal@weizmann.ac.il

The human gut is the richest ecosystem of microbes in the human body, being composed of an estimated $10^{13}$ microbial cells[1]. Each person's gut harbors few hundreds of species out of the thousands which can occupy this ecological niche. Many of these species are difficult to cultivate despite an ever-growing effort based on new culturing methods. In addition, a grand effort has taken place in past years to improve upon assembling new microbial genomes from metagenomic sequencing, and to establish techniques of classifying these genomes to create more elaborated reference sets of the human microbiome.

The gut microbiome is in constant cross-talk with the cells and systems of our body, and is linked to human health and disease[2,3]. Human health both affects and is affected by gut microbiome composition, at the level of richness and diversity of the microbiome, the influence of dysbiosis, and even the existence and abundance of specific microbes[4,5]. Furthermore, microbial genes and variations within those genes have shown to affect the potency of drugs, a research subject in its infancy[6].

Our understanding of the microbes residing within our bodies and their effect on our health is limited by our ability to identify the microbial content. The two main paths to microbial research are either through marker genes, such as the 16S ribosomal RNA genes, or through reference alignment of shotgun read sequencing. In both cases, the broad genetic contents of each microbe is deduced from what is known about its genome. Continual efforts are made to expand this knowledge by establishing better quality and more comprehensive genomes, by using innovative methods and more samples gathered from different populations.

In this work, we build upon previously published genomes and our own set of assembled genomes (aka assemblies) from 51,052 metagenomic samples, and apply strict quality control and anti-biasing considerations to form a set of 241,118 genomes. We proceed to cluster this set, based on the genomic distances between pairs, and choose a representative genome for each cluster. In this way, we construct a genome reference set with median completeness of 95% and median contamination of 0.67%, which represents 3594 gut microbial species. All but 17 of these species are from the Bacteria domain, with the rest being Archaea. We term our representative set of genomes the Weizmann Institute of Science or "WIS" reference set.

We show our reference set recapitulates more reads of the gut microbiome than the previous benchmark in the field, using validation cohorts representing both western and non-western populations. Overall, our reference set introduces 310 novel species, and highlights others that were not known to exist in the human body. Thus, our reference set contributes to the grand effort of expanding the knowledge of the human microbiome.

## Results

**Building a new reference set.** Our process of creating the human gut microbial genomes reference set follows the work in ref. ("Methods")[7]. Their work used 9428 metagenomic samples from multiple human body sites, although mostly from the gut, to recapitulate 4930 microbial species, many of which were not known before (Table 1 and Fig. 1). From here after we term their set of genomes the University of Trento or "UNITN" reference set. We will later refer to a second set, the Unified Human Gastrointestinal Genome collection, as the "UHGG" reference set[8].

We sought to improve the previously published set, by utilizing 51,052 human gut microbiome samples from various cohorts, 6 times more samples than were used in ref. [7]. These cohorts include individuals with a variety of diseases such as diabetes, multiple sclerosis, cardiometabolic syndromes, fatty liver, cancer, and irritable bowel disease[9-14]. Other than allowing to choose

**Table 1 Main characteristics of the datasets used to construct the WIS, UNITN, and UHGG reference sets.**

| | N | WIS | UNITN | UHGG |
|---|---|---|---|---|
| **Samples** | Samples | 51,052 | 9428 | Did not generate new assemblies from samples |
| | Body sites | 1: Gut (100%) | 5: Gut (85%), oral (8.5%), skin (5.4%), vagina (1%), and maternal milk (0.1%) | NA |
| | Countries | 2: Israel (90%) and USA (10%) | 31: USA (15%), China (14%), Israel (10%), Sweden (6%), and Denmark (6%)* | NA |
| | Age | Adults (99%) and children (<1%) | Adults (81%) and children (19%) | NA |
| | Gender | Female (61%) and male (39%) | Not specified | NA |
| **Assemblies** | Assemblies from samples before filtration criteria | 483,192 | 345,654 | 0 |
| | Assemblies from samples after filtration criteria# | 142,912 (30%) | 154,723 (45%) | 0 (0%) |
| | External assemblies after filtration criteria# | 98,206 (88% Passoli et al.[7]) | 80,990 | 286,997 (48% Passoli et al.[7]) |
| | Total assemblies used | 241,118 (36% Passoli et al.[7]) | 154,723 | 286,997 (48% Passoli et al.[7]) |
| **Clusters** | Species | 3594 | 4930 | 4644 |
| | Genera | 2365 | 2640 | Not specified |
| | Families | 627 | 778 | Not specified |

*Five most represented countries.
#Each set of assemblies went through different filtration criteria ("Methods").

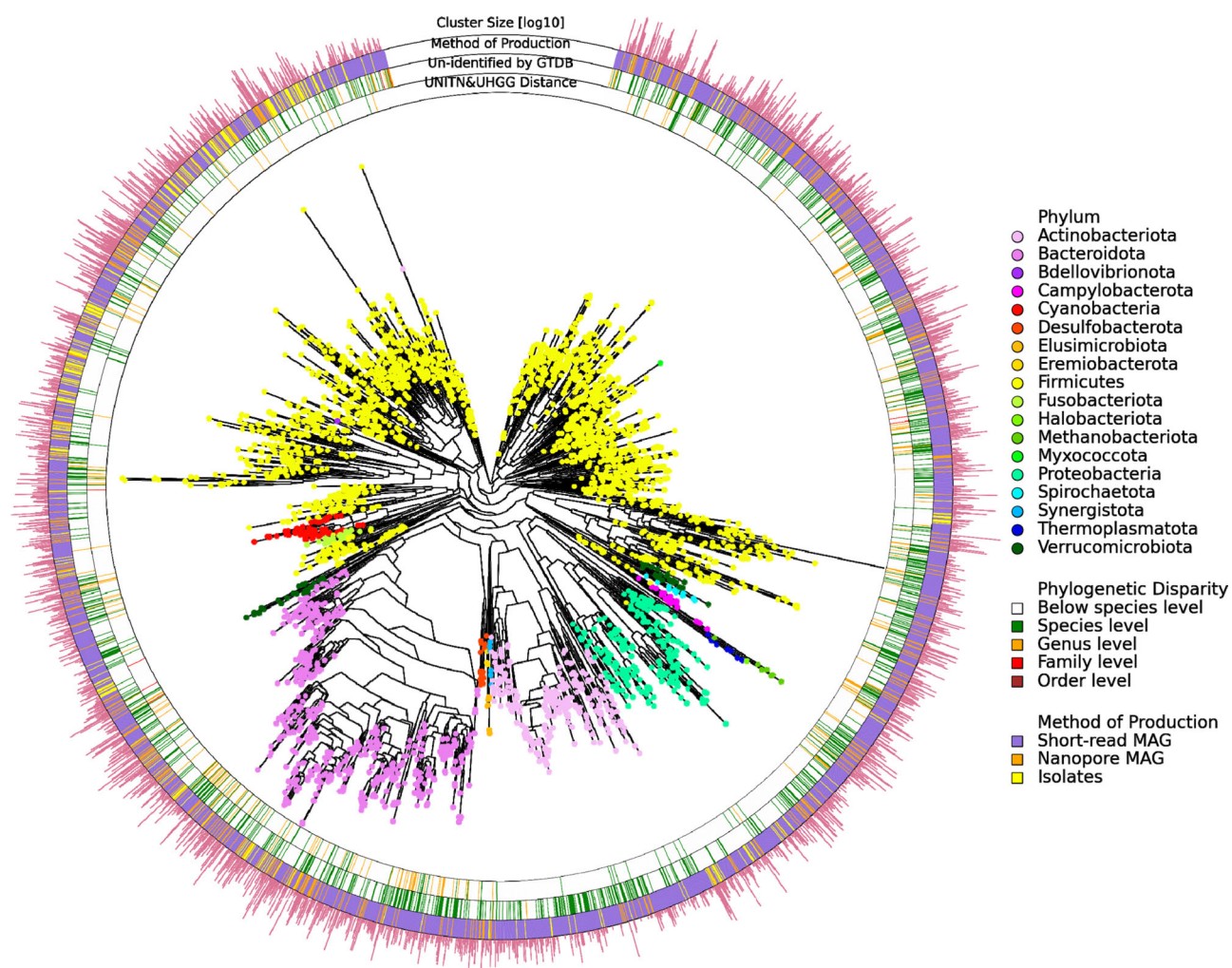

**Fig. 1 Phylogenetic tree of the new human gut microbiome genome reference set.** Inside the circle is a phylogenetic tree of the WIS representative genomes colored according to their Genome Taxonomy Database (GTDB) assigned phylum (see color legend). Inner ring shows the distance each of the WIS genomes has from the closest genome in the UNITN or UHGG reference sets, stratified by its phylogenetic distance level—below species level in white, species level in green, genus level in orange, family level in red and order level in brown. Second ring (counting from the inside) shows the highest phylogenetic level the genome was un-identified at, by GTDB—same colors as in the inner ring. Third ring shows the genome production method—short-read metagenome-assembled genomes (MAG) in purple, nanopore MAG in orange, and isolates in yellow. Outer ring line length is proportional to the $\log_{10}$ of the cluster size each genome represents.

higher-quality representative genomes, having more samples from different individuals elevates the chances of assembling rare microbial species' genomes, that in most people exist in lower abundance than is required for the genome assembling process.

Our metagenomic samples are mostly of Israeli adults (range 4–93 years old, median 55) but we also complimented the 142,912 newly assembled genomes produced from our samples with assemblies we gathered and curated from previously published studies (up to 2019) originating in different countries (short-read metagenome sequencing-based assemblies: 86,191 from ref. [7] and 7211 from ref. [15], isolate-based assemblies: 94 from ref. [16], 1327 from ref. [17], and 528 from ref. [18] and ref. [19], and nanopore metagenome sequencing-based assemblies: 2855) (Table 1 and "Methods")[7,15–19]. Metadata of all 241,118 assemblies used is available in Supplementary Data 1. We chose not to include public repositories, such as the National Center for Biotechnology Information (NCBI) genomes for two main reasons. First, as it has no metadata which is required for our filtration criteria ("Methods") and second, as it is biased toward species that are in the focus of scientific research. For example, *Escherichia coli* is a

relatively low abundant species in the human gut, however, it portrays 10% of the "NCBI-RefSeq" dataset as of December 2020, while *Prevotella copri* is a highly abundant species in the human gut but it is represented by less than 0.05% of the dataset. Having biases between the number of different species' assemblies, unrelated to their actual prevalence or abundance, can harm the clustering process that aims to group assemblies from different sources by the species they belong to. Naturally, these over-represented species, found in public repositories, will also be assembled from metagenomic samples or from isolates. Due to the same bias concern, we chose to focus on a single body site, the human gut (Table 1 and "Methods").

**Comparison to UNITN.** In comparison to UNITN our new reference set is composed of genomes that are of significantly higher quality (median completeness 95% vs. 91%, $q < 0.0001$ ($q$ is the Bonferroni corrected $P$ value) and median contamination 0.67% vs. 0.70%, $q < 0.05$, Mann–Whitney $U$ test), have significantly longer contigs (measured by N50, half of the genome sequence is covered by contigs larger than or equal to the N50

contig size) (median N50 $5.57e + 04$ base pairs (bp) vs. $3.04e + 04$ bp, $q < 0.0001$, Mann–Whitney $U$ test), have overall significantly higher genome length (median length $2.22e + 06$ bp vs. $1.96e + 06$ bp, $q < 0.0001$, Mann–Whitney $U$ test) and represent bigger clusters (mean cluster size 67.09 vs. 39.61, $q > 0.05$, Mann–Whitney $U$ test) (Fig. 2a–d, g). For the same information regarding all assemblies used (rather than just the ones that were chosen to be the species' representatives) see Supplementary Fig. 1.

The notable difference in completeness is not incidental as we set this parameter threshold to be 70% for all of our assemblies while Passoli et al. chose a threshold of 50%[7]. We chose to increase the completeness threshold, as higher completeness results in lower systematic bias in quality parameters estimates, and this bias is very small (<2%) in cases where genomes are over 70% complete and less than 5% contaminated[20–22]. Higher completeness also ensures clustering together of partial assemblies of the same genome ("Methods"). The notable difference in contigs length expressed by the N50 parameter is driven mostly by the alternative methods for curating genomes we used, nanopore and isolates as will be later shown (Fig. 1, third ring). The WIS reference set has fewer genomes (3594 vs. 4930), which is expected given the single body site used to create it. For example, species that are exclusive to the skin microbiome should not appear in our gut-derived reference set (Table 1).

Computational approaches of producing genomes can suffer from mis-binning, the association of unrelated contigs resulting in chimeric genomes. In order to mass evaluate the frequency of this potential issue, we used Genome UNClutterer (GUNC) which assesses the probability of a genome being a chimeric mixture of distinct lineages by using a clade separation score (CSS) ("Methods")[23]. We found that in comparison to UNITN our reference has significantly fewer genomes suspected to be chimeric (6.82% vs. 11.36% CSS > 0.45, median CSS 0.09 vs. 0.12, $q < 0.0001$, Mann–Whitney $U$ test) (Fig. 2e).

We procured fewer assemblies per sample than UNITN for several reasons. Our samples are all single-end with $9.74e + 08 \pm 3.18e + 08$ bp per sample (computed by read length times read depth), while UNITN comprises both single-end and paired-end samples with $4.21e + 09 \pm 3.39e + 09$ bp per sample, this lower number of bp restricts the number of assemblies that can be produced from a sample with current computational tools. Furthermore, applying harsher quality control and filtration criteria reduced the number of assemblies we obtained from each sample.

A principal issue with the UNITN reference set is that a few of the phylogenetic species in it are represented by multiple genomes, even though both reference sets' clusters were created by the same hierarchical process bounded by an average MinHash (MASH) distance of 0.05 (a proxy for one minus the accepted species-level boundary of 95% average nucleotide identity (ANI)) ("Methods")[24,25]. For example, according to the UNITN reference set, there are 15 genomes that belong to the species *Clostridium sp*, 13 to the species *Streptococcus mitis* and 188 additional genomes that have a non-unique species identification. In addition, the closest pair of genomes in it is 0.035 MASH distance apart. Due to the nature of how species were historically delineated based on biochemical properties and morphology, non-unique species identification is not an issue in itself. However, the genomic similarity between representative genomes exemplified by the MASH distance is a problem since it causes inconclusive read alignment. Our strict quality control, anti-biasing considerations and focus on the human gut microbiome resulted in only 79 genomes with non-unique species identity and none of the taxonomic species names were assigned to more than seven genomes. Moreover, the closest pair of genomes in our reference set is 0.040 MASH distance apart (Fig. 1 outer ring and Fig. 2h).

Passoli et al. termed 11,402 species clusters as non-human as they included only NCBI genomes and no assemblies from any of their samples[7]. Notably, 77 of our genomes are assigned to these "non-human" species clusters, and given that all our assemblies are from human stool this suggests these 77 species do exist in humans after all.

In order to evaluate how well our new reference set recapitulates reads from short-read metagenomic samples in comparison to this previous benchmark, we aligned an external source of 1528 samples of individuals from the Netherlands that participated in the LifeLines study onto both reference sets ("Methods", Fig. 3a, and Supplementary Data 2)[26,27]. The percentage of reads that align to a reference set with no more than a few errors or as commonly called "mapped", is significantly higher in the WIS reference set than in the UNITN set (median value of 83.65% vs. 79.78%, $q < 0.0001$, Mann–Whitney $U$ test). This reduces the amount of unknown matter, expressed by un-aligned reads, by 19.1% with some of the remaining DNA content not even expected to be covered by any of these reference sets that do not include viruses, fungi, and food.

Unique alignment, portrayed by the percentage of reads that are better aligned (in terms of mapping errors) to a single area in one representative genome of the reference set than to any other, is a sign of increased variability within the reference set. Therefore, we also measured this parameter and found that the gap in unique read alignment between the sets is larger than the gap in non-unique alignment (median value of 65.20% vs. 58.51%, $q < 0.0001$, Mann–Whitney $U$ test).

We conducted the same process on 3096 held-out samples from our own cohort where we received similar results, and on three smaller cohorts of non-western populations from India ($n = 110$), El Salvador ($n = 113$), and Tanzania ($n = 68$), that are often less represented in scientific research[28–30]. On the non-western cohorts, the difference in read alignment was not significant, and in one case, of the Tanzanian cohort, it was even significantly higher in UNITN, perhaps as Tanzania is relatively highly represented in the UNITN set, and did not necessarily make it into our set. In terms of unique alignment, the WIS significant advantage was maintained throughout the non-western populations tested, which shows our process was broad enough and did not specifically capitulate the typical Israeli adult gut microbiome (Fig. 3b–e).

Accounting for more of the genetic content, as exemplified by the increased mappability, aids in studying the microbes. In particular, it helps in understanding their genetics and the effect variations within their genomes have on their capabilities and on host health.

**Different methods of producing genomes**. In order to produce this improved quality and more divergent reference set, we complemented the short-read devised metagenome-assembled genomes (MAG) with data from nanopore sequencing and sequencing of isolates, both methods produce higher-quality assemblies[16–19]. These alternative method assemblies represent 2% of our assembled genomes and 21% of our chosen species representative genomes. The 21% are composed of 12% nanopore-based MAG that were produced from a small subset of our samples and 9% isolates' based genomes that were publicly available.

These two methods produce genomes that are significantly improved in all quality measures (completeness, contamination, N50, and genome length) to the short-read based MAG ($q < 0.0001$, Mann–Whitney $U$ test) (Fig. 4a–d). In our reference set, species representative genomes from isolates significantly outperformed nanopore-based MAG in terms of completeness

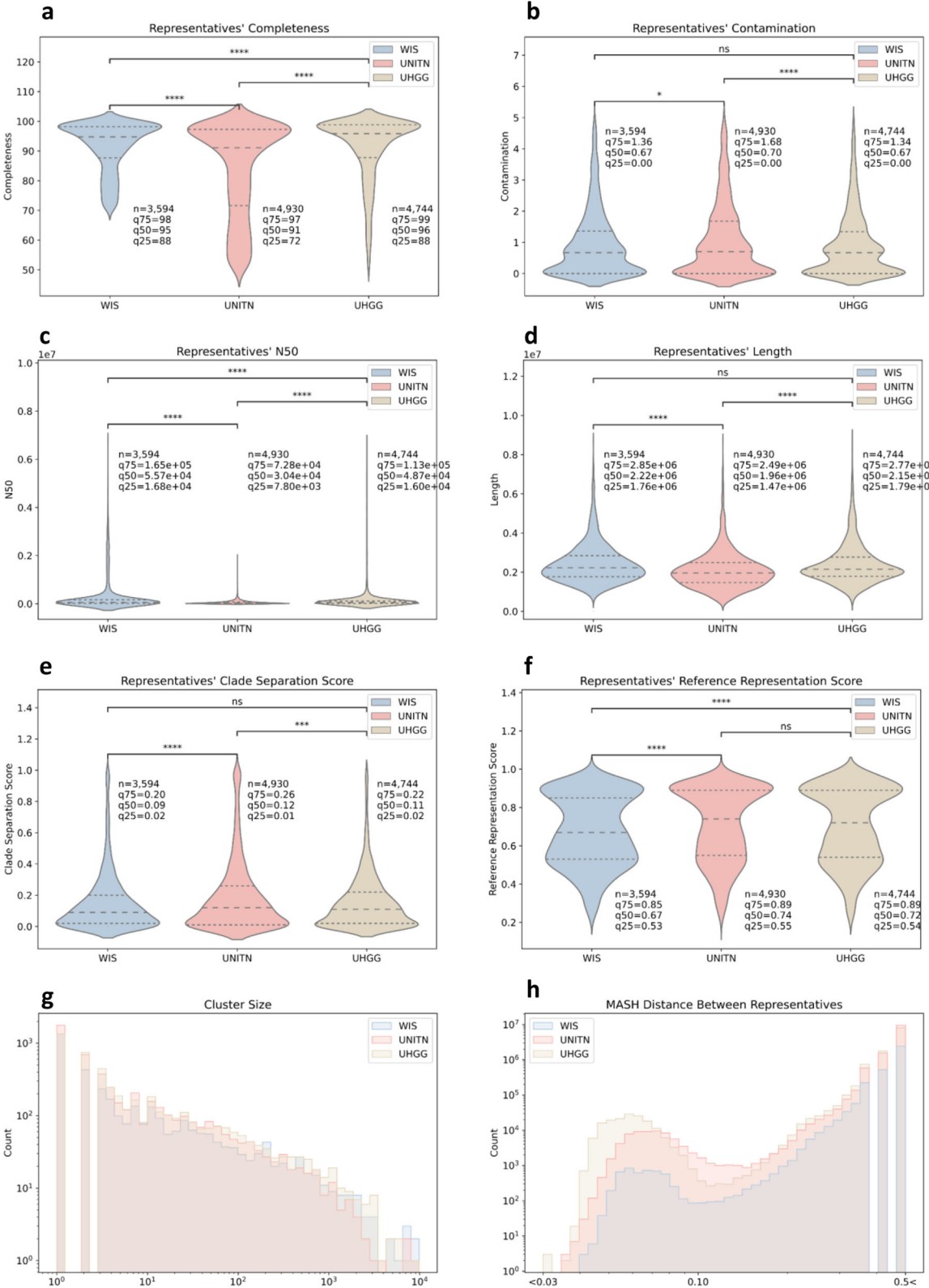

**Fig. 2 The WIS reference set shows higher-quality measures than UNITN.** WIS ($n = 3594$ genomes) in blue, UNITN ($n = 4930$) in red and UHGG ($n = 4744$) in Brown. **a–f** are violin plots of the completeness, contamination, N50, length, CSS, and RRS of the representatives genomes. In each plot, the $y$ axis is the parameter value and drawing width is the kernel density estimate. The dashed line is the median and the dotted lines are the interquartile range. **g** Cluster size histogram. **h** MinHash (MASH) distance between representatives histogram. Both axes are in $\log_{10}$ scale. Bonferroni corrected $P$ value annotations: not significant (ns) $q > 0.05$, *$q < 0.05$, **$q < 0.01$, ***$q < 0.001$, ****$q < 0.0001$ according to Mann–Whitney $U$ test. The completeness threshold in WIS was 70% while it was 50% in UNITN and UHGG. Panels b, c and e seemingly have values under zero, this is a graphical illusion of violin plots since the distribution is dense around the minimal value (zero).

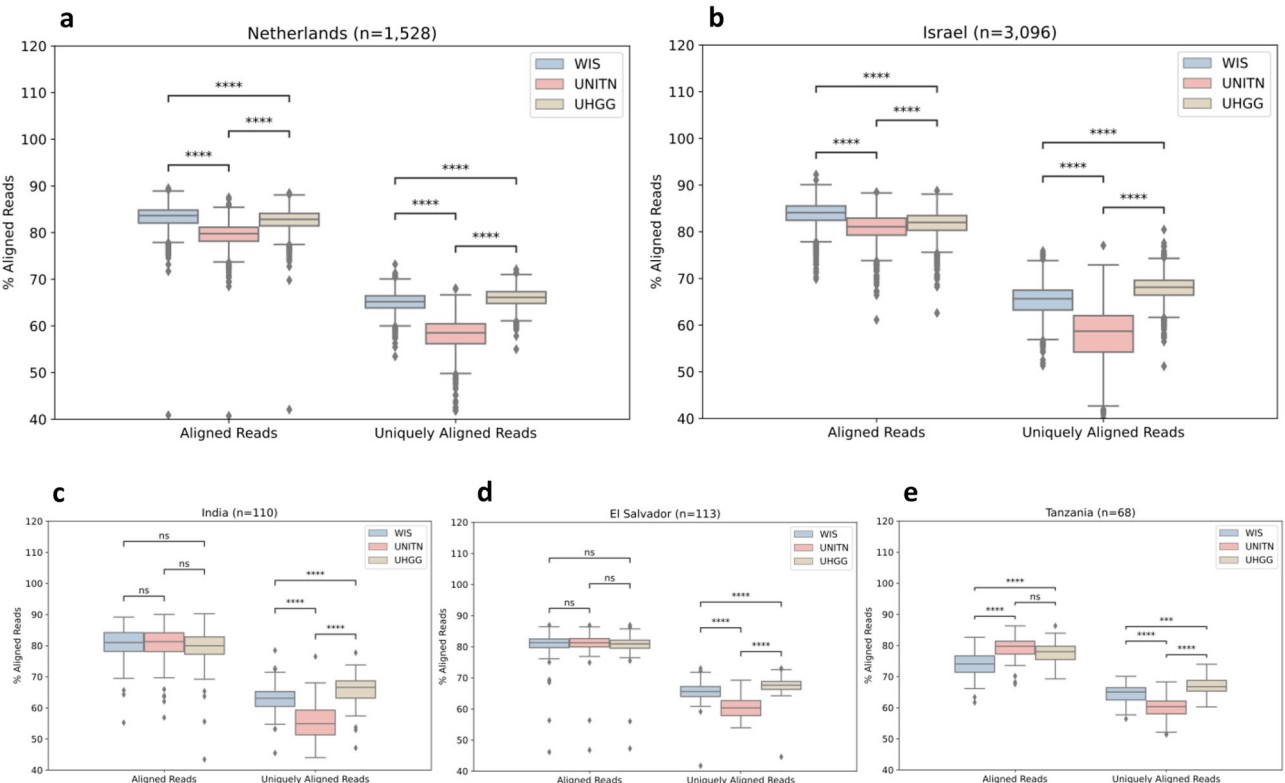

**Fig. 3 More reads are successfully aligned to the WIS reference set than to the UNITN reference set.** WIS in blue, UNITN in red and UHGG in brown. **a** Netherlands (*n* = 1528 samples), **b** Israel (*n* = 3096 held-out samples), **c** India (*n* = 110), **d** El Salvador (*n* = 113), and **e** Tanzania (*n* = 68) percentage of reads (*y* axis) that were aligned (left) and that were uniquely aligned (right) to each genome reference set. The boxes show the quartiles of the dataset while the whiskers extend to 1.5 of the interquartile range, points beyond the whiskers are considered to be outliers. Bonferroni corrected *P* value annotations: not significant (ns) *q* > 0.05, \**q* < 0.05, \*\**q* < 0.01, \*\*\**q* < 0.001, \*\*\*\**q* < 0.0001 according to Mann–Whitney *U* test.

(median value of 99 vs. 96, q < 0.0001, Mann–Whitney *U* test) and genome length (median value of 3.37e + 06 vs. 2.66e + 06, q < 0.0001, Mann–Whitney *U* test), but were significantly lower than the nanopore genomes in terms of N50 (median value of 2.16e + 05 vs. 1.88e + 06, *q* < 0.0001, Mann–Whitney *U* test). With regards to contamination there was no significant difference between the two methods (median value of 0.24 in isolates vs. 0.40 in nanopore-based MAG, *q* > 0.05, Mann–Whitney *U* test). For the same information regarding all assemblies used (rather than just the ones that were chosen to be the species' representatives) see Supplementary Fig. 2.

In terms of CSS, nanopore MAG have significantly fewer genomes suspected to be chimeric than isolates, and both have a significantly lower chance than short-read MAG (0.88% of nanopore MAG, 1.28% of isolates and 8.39% of short-read MAG have CSS > 0.45, median CSS 0.00, 0.07, and 0.12, respectively, *q* < 0.0001, Mann–Whitney *U* test) (Fig. 4e).

Despite these two alternative sources for genome assembly having many advantages, they are still more expensive and thus can only be applied to a small number of samples. Furthermore, not all gut microbes have yet been successfully cultured. Therefore, even though we enriched our genome pool with these improved quality genomes, most of our assembled genomes (98%) and the chosen species' representative genomes (79%) are still produced from short-read based MAG.

**Genetic annotations**. Using standard tools we annotated the genomes in both the UNITN and WIS reference sets ("Methods", Supplementary Data 3). The total number of segments identified as genes, not necessarily annotated, is 8,138,394 in WIS and 9,506,317 in UNITN. The absolute number of uniquely annotated genes, that

is counting each gene once even if it appears in multiple species, is 30,545 in WIS and 30,730 in UNITN. These numbers translate to 17.18% more uniquely annotated genes per bp in the WIS set than in the UNITN set ($3.5 \times 10^{-6}$ vs. $3.0 \times 10^{-6}$), 12.49% more uniquely annotated gene products per bp ($1.4 \times 10^{-6}$ vs. $1.2 \times 10^{-6}$) and 10.47% more uniquely annotated enzymes per bp ($5.2 \times 10^{-7}$ vs. $4.7 \times 10^{-7}$). These results were unexpected since the UNITN reference set includes genomes that originated from multiple body sites (although most are from the gut) and thus are expected to cover a wider range of functions while the WIS reference set originated only in a single body site—the gut. A possible explanation for these results is that the increased quality of the WIS genomes made them more accurate and thus it was easier for the annotation tools to recognize the genes in the reference set. Another possible complementary explanation is that the higher number of starting samples allowed the curation of genomes and hence functions of microbes that are missing from the UNITN reference set. In contrast, this could stem from the gut being a more researched body site, and hence more annotated. Regardless of the reason, since the absolute number of uniquely annotated genes is comparable this indicates that, specifically for the gut microbiome, our new reference set covers a wider range of genes and hence functions than the UNITN reference set.

We broke down the annotated segments into the following categories—protein-coding (CDS), transfer ribonucleic acid RNA (tRNA), miscellaneous RNA (miscRNA), ribosomal RNA (rRNA), transfer-messenger RNA (tmRNA), and repeating regions (Fig. 5). In all categories, the WIS reference set had significantly more annotations per genome than UNITN (*q* < 0.0001, Mann–Whitney *U* test). The gap between the WIS and UNITN reference sets is most prominent in the rRNA category where there are almost five times more occurrences per genome. Deepening into this category, we

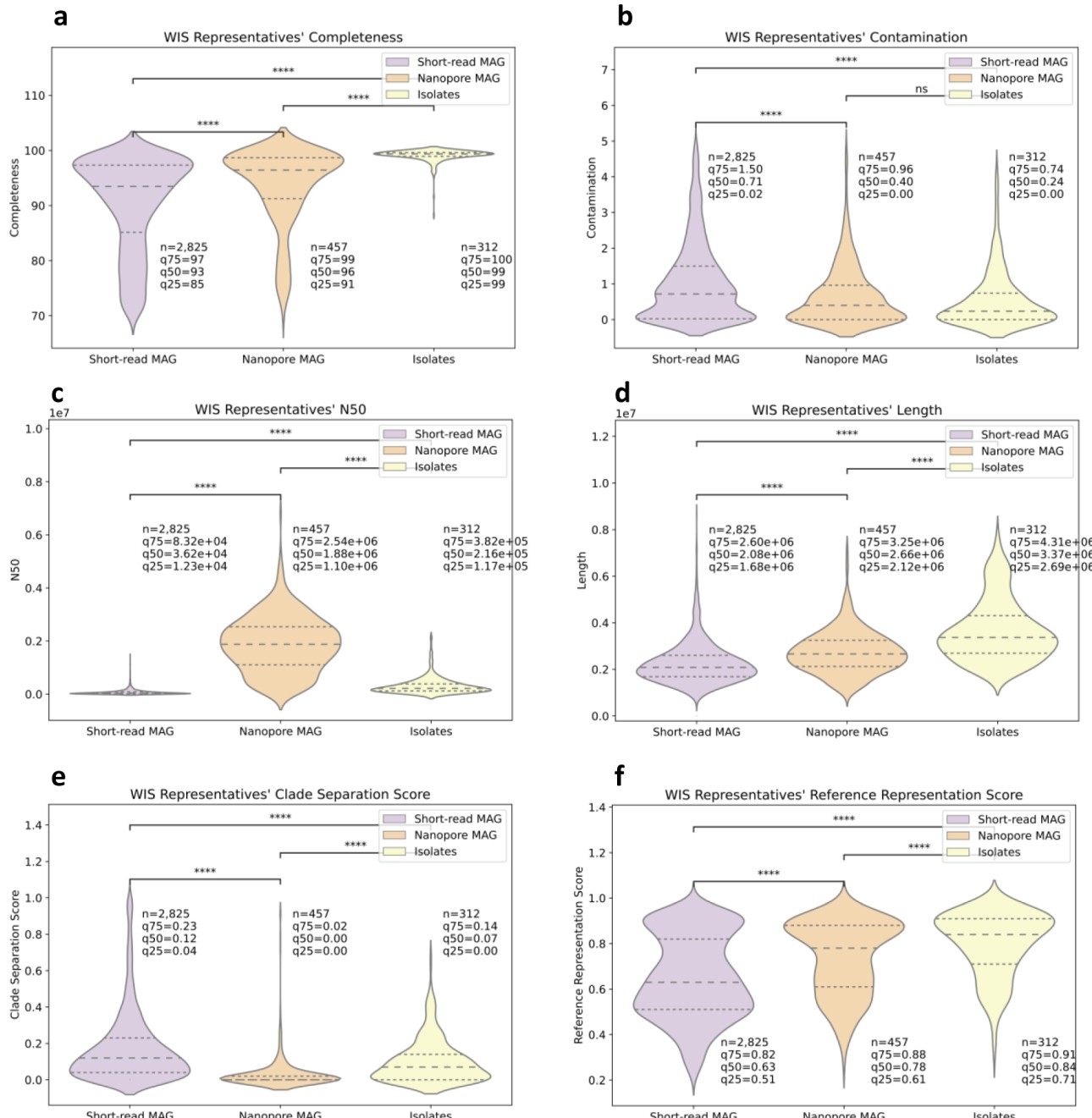

**Fig. 4 Nanopore and isolates representative genomes show higher-quality measures than those of short-read-based MAG.** Short-read metagenome-assembled genome (MAG) ($n = 2825$ genomes) in purple, nanopore MAG ($n = 457$) in orange and isolates ($n = 312$) in yellow. **a**–**f** are violin plots of the completeness, contamination, N50, length, CSS, and RRS of the representatives genomes, respectively. In each plot, the $y$ axis is the parameter value and drawing width is the kernel density estimate. The dashed line is the median and the dotted lines are the interquartile range. Bonferroni corrected $P$ value annotations: not significant (ns) $q > 0.05$, *$q < 0.05$, **$q < 0.01$, ***$q < 0.001$, ****$q < 0.0001$ according to Mann–Whitney $U$ test. Panels b, c, and e seemingly have values under zero, this is a graphical illusion of violin plots since the distribution is dense around the minimal value (zero).

found that there are 1.04 5S genes, 0.68 16S genes and 0.63 23S genes per genome on average in the WIS reference set, while there are only 0.35 5S, 0.10 16S, and 0.06 23S genes per genome on average in the UNITN reference set—a 2.97, 6.72, and 10.15-fold differences, respectively (all significantly different, $q < 0.0001$, Mann–Whitney $U$ test). This increased rRNA curation is highly driven by the alternative methods of genome assembly—isolates and even more so nanopore. The 16S gene is of particular interest as it is known to be present in all bacteria and consists of a highly conserved region that can be identified using PCR, inter-spread by

variable regions that are species or genus specific[31]. The downside of identifying bacteria using 16S is that it requires prior knowledge to tie the variable 16S regions' sequences to a taxonomy, and that it only recognizes bacteria presence and not its full genomic context. For many of the gut bacteria, this prior knowledge does not exist but can now be deduced using our expanded reference set.

**Comparison to UHGG**. A new reference set, expanding the gut segment of UNITN in ref. [7], was recently published as the Unified

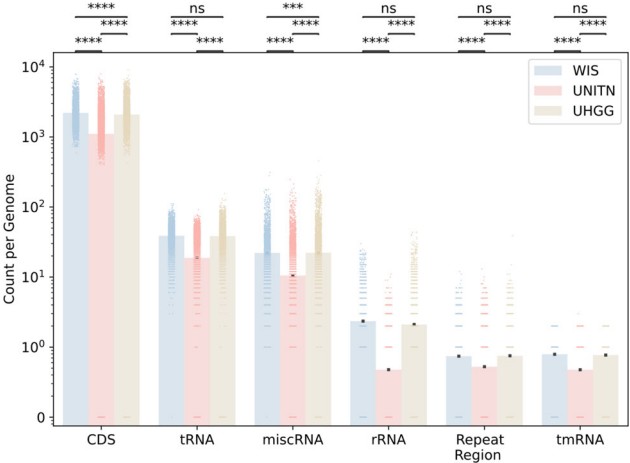

**Fig. 5 The WIS reference set is more annotated than the UNITN reference set.** Count per genome (each genome is a dot and the mean overall genomes is shown as a bar, y axis is in log₁₀ scale) of each type of annotated segment (x axis). Protein coding (CDS), transfer ribonucleic acid RNA (tRNA), miscellaneous RNA (miscRNA), ribosomal RNA (rRNA), transfer-messenger RNA (tmRNA), and repeating regions, stratified by reference set— WIS ($n = 3594$ genomes) in blue, UNITN ($n = 4930$) in red, and UHGG ($n = 4744$) in brown. Bonferroni corrected $P$ value annotations: not significant (ns) $q > 0.05$, *$q < 0.05$, **$q < 0.01$, ***$q < 0.001$, ****$q < 0.0001$ according to Mann–Whitney $U$ test.

Human Gastrointestinal Genome collection[8]. This reference set has not yet been established as a benchmark, but a thorough comparison to our work is still required. Their assembly set does not introduce new data, and overlaps with ours in many of the public datasets used, but it also includes genomes from NCBI, PATRIC, and IMG repositories that we find to be biased towards species that have been the focus of scientific research. The UHGG assembly set is composed of 286,997 genomes, which by their criteria (of 99.9% ANI between two assemblies from the same sample) are considered to be 204,938 non-redundant genomes, 15% less than the size of our assemblies set, where our anti-biasing considerations inherently excluded assemblies from the same person in cases where they were under 0.05 MASH distance apart (equivalent to being over 95% ANI apart) (Table 1).

The UHGG assembly set was derived by using lower quality thresholds (completeness >50% and contamination <5%, combined with an estimated quality score (completeness-5*contamination) >50), and was iteratively clustered and later combined to a resulting set of 4644 species-level clusters, 29% more species than in the WIS reference set. The same quality parameters were considered for choosing representatives in both UHGG and WIS, with only the N50 weight being larger in WIS. However, in UHGG, isolates were prioritized over MAG, that is, even if a MAG had a higher-quality score than an isolate genome, the latter would still be chosen as the representative.

In terms of quality, the UHGG representative genomes are significantly more complete than the WIS representative genomes (median completeness 96% vs. 95%, $q < 0.0001$, Mann–Whitney $U$ test) but have significantly shorter contigs (median N50 4.87e + 04 bp vs. 5.57e + 04 bp, $q < 0.0001$, Mann–Whitney $U$ test). With regards to contamination and genome length there is no significant difference between the two sets (median contamination 0.67% vs. 0.67%, $q > 0.05$ and median length 2.15e + 06 bp vs. 2.22e + 06 bp, $q > 0.05$, Mann–Whitney $U$ test) (Fig. 2a–d). Perhaps their higher completeness, even while using lower quality thresholds, stems from the prioritization of isolate genomes over MAG, which as we showed tend to be more

complete. For the same quality information regarding all assemblies used (rather than just the ones that were chosen to be the species' representatives), see Supplementary Fig. 1.

UHGG has significantly smaller clusters (mean cluster size 60.97 vs. 67.09, $q < 0.0001$, Mann–Whitney $U$ test), and more genomic similarity among its representative genomes in comparison to WIS (0.019 vs. 0.040 minimal MASH distance) (Fig. 2g, h). Taken together, this might indicate redundancies in the set of representatives, possibly due to the iterative clustering process. There is no significant difference in the likelihood of the representative genomes between the two sets to be chimeric (6.82% vs. 5.88% CSS > 0.45, median CSS 0.09 vs. 0.11, $q > 0.05$, Mann–Whitney $U$ test) (Fig. 2e).

In terms of the percentage of read alignment of the external validation cohort, the UHGG reference set recapitulates significantly fewer reads than the WIS reference set (median value of 82.85% vs. 83.65%, $q < 0.0001$, Mann–Whitney $U$ test), but in terms of unique read alignment UHGG recapitulates significantly more reads than WIS (median value of 66.12% vs. 65.20%, $q < 0.0001$, Mann–Whitney $U$ test) (Fig. 3a). We obtained similar results on the internal validation cohort. However, as in UNITN, on the smaller non-western cohorts the difference in read alignment was not significant, and in one case, of the Tanzanian cohort, it was even significantly higher in UHGG. In terms of unique alignment, UHGG maintains its significant advantage over WIS (Fig. 3b–e).

There are 1158 genomes in the UHGG reference set which are not under the species-level genomic distance from any WIS representative, and there are 406 species in WIS that are not in the UHGG reference set by the same criteria. Creating a combined reference set of all the WIS representatives with the additional 1158 representative genomes from UHGG recapitulates a median value of 84.43% of the reads, significantly higher than either one of the two reference sets on their own ($q < 0.0001$, Mann–Whitney $U$ test). However, this combined set is significantly worse in terms of unique read alignment than each of the reference sets on their own, as it only captures a median value of 64.9% of the reads ($q < 0.0001$, Mann–Whitney $U$ test). Evaluating the prevalence of the unique species of each set, we found the UHGG unique species have significantly lower prevalence than the WIS unique species (mean prevalence in the external validation cohort of 11.51% vs. 26.14%, $P < 0.01$, and in the internal validation cohort of 14.09% vs. 36.29%, $P < 0.0001$, Mann–Whitney $U$ test).

In order to evaluate the difference in annotations, we applied the same standard tools to the UHGG as we did to the WIS reference set ("Methods"). The total number of segments identified as genes, not necessarily annotated, is 8,138,394 in WIS and 10,547,568 in UHGG. The absolute number of uniquely annotated genes is 30,545 in WIS and 33,376 in UHGG. Due to the higher number of genomes in the UHGG set, these numbers translate to 19.71% more uniquely annotated genes per bp in the WIS set than in the UHGG set ($3.5 \times 10^{-6}$ vs. $2.9 \times 10^{-6}$), 19.01% more uniquely annotated gene products per bp ($1.4 \times 10^{-6}$ vs. $1.2 \times 10^{-6}$) and 20.51% more uniquely annotated enzymes per bp ($5.2 \times 10^{-7}$ vs. $4.3 \times 10^{-7}$). When breaking down the annotated segments to categories, the WIS reference set had significantly more annotations per genome than UHGG in the CDS and miscRNA categories ($q < 0.0001$ and $q < 0.001$, respectively, Mann–Whitney $U$ test), in the rest of the categories, there was no significant difference between the reference sets (Fig. 5).

As our work is based on highly curated assemblies, introduces 142,912 assembled genomes and includes 406 species that are not represented by UHGG, it expands the knowledge of the human gut microbiome. In the future, both the UHGG and WIS sets, together with other works that will follow, should be combined

into a more comprehensive genome set that considers the different methodologies and better represents a variety of populations. Meanwhile, it is possible to add the UHGG species not covered by WIS to our reference set, with the advantages and disadvantages it brings.

**Newly discovered human gut microbial species.** We evaluated the contribution of our reference set in two ways. First, in order to understand whether the microbes our reference genomes represent are already known, we identified the genomes using the Genome Taxonomy Database (GTDB) ("Methods")[32]. Second, in order to understand if the microbes our genomes represent were already found by the UNITN or UHGG sets, we measured the distance each of the WIS reference set genomes has to the closest UNITN or UHGG reference genome ("Methods").

We found that 1055 genomes (out of the 3594 we have) were not identified as a known species by GTDB-Tk, a toolkit to classify genomes using GTDB[32]. Of those unknown genomes, 93 were not classified at the genus level, two were not classified at the family level and one was not even classified at the phylogenetic order level (Fig. 1, second ring and Fig. 6a).

When comparing the WIS genomes to the UNITN and UHGG sets, we found that 340 genomes are so genetically far from either reference sets that they surpass the genomic distance that defines a species and hence should be considered as an independent microbial species (Fig. 1, inner ring and Fig. 6b)[25]. Similarly, 198 out of the 340 genomes surpassed the genus level distance and six even surpassed the family-level distance and thus should be considered as unfamiliar families relative to the UNITN and UHGG reference sets.

Integrating the two outlooks together, i.e., which microbes are known and which microbial genomes were already included in the other reference sets, we found that 310 of our genomes belong to microbial species that were not described before, 19 of which belong to unfamiliar genera (Fig. 6c). The novelty of these species is further supported by the GUNC reference representation score (RRS), an estimate of how closely a query genome is already represented by the GUNC reference set (unknown species would have RRS close to 0, while known ones will have RRS closer to 1). The RRS is significantly lower for the WIS representatives than for UNITN and UHGG representatives (median RRS 0.67 vs. 0.74 and 0.72, respectively, $q < 0.0001$, Mann–Whitney $U$ test) (Fig. 2f), and is much lower in the 310 novel species compared to the known ones (median RRS 0.58 vs. 0.68, $P < 0.0001$, Mann–Whitney $U$ test).

These 310 newly discovered species are mostly singletons (76%), meaning they were the only genome in their species-level cluster, even though some do exist in many individuals in relatively low abundance. These species representative genomes often originated from our own samples (98%) and are usually short-read MAG (95%). As these species are mostly MAG singletons, they are more likely to be suspected chimeras (11.61% of the novel species vs. 6.36% of the known ones have CSS > 0.45, median CSS 0.16 vs. 0.09, $P < 0.0001$, Mann–Whitney $U$ test), close to the likelihood of the UNITN species (11.36% CSS > 0.45).

For the biggest phylum in the human gut microbiome, Firmicutes, we found 156 undescribed species that constitute 7% of the number of genomes it holds in the WIS reference set. Similarly, for the second largest phylum in the human gut, Bacteroidota, we found 98 undescribed species that constitute 17% of its genomes in the WIS reference set. Other smaller phyla, such as Spirochaetota and Eremiobacterota, include only a few undescribed species but those represent over 20% of the genomes we have for them (Supplementary Fig. 3). Three phyla have a single genome in the WIS reference set and of them, one was

found to be an undescribed species of the Halobacteriota phylum, Methanomicrobia class (Supplementary Fig. 3 and Fig. 6d). This species is of the archaeal domain and *Methanocorpusculum* genus, and it originated from a singleton short-read MAG of our own samples.

The Kiritimatiellae class is highly expanded by our newly discovered species which constitute 80% of all its genomes in the WIS reference set (Fig. 6d). By GTDB identifiers, we know that the species of this class in the WIS reference set belong to two unnamed genera. One genus has a single genome in it and it is of a known human gut species that was also represented in the UNITN and UHGG reference sets. The other genus has four species, which are all new and stem from clusters with one or two genomes in them, all these genomes are short-read MAG from our own samples. These four species are within genus distance level from the closest UNITN and UHGG genomes. The WIS Kiritimatiellae reference genomes are 0.14–0.34 MASH distance apart from each other.

The most prevalent new species in both the internal (15.02% of the samples had it with a mean relative abundance among those who had it of 0.10%, and a maximal relative abundance of 1.02%) and external (18.98% of the samples had it with a mean relative abundance among those who had it of 0.11%, and a maximal relative abundance of 0.85%) validation cohorts is a species of the genus *Faecalibacterium* ("Methods" and Supplementary Data 4). It is a singleton short-read MAG from ref. [15]. Another new species of this genus is third in its prevalence in the internal validation, and second in the external validation cohort.

Among those who had it, the most abundant novel species in the internal validation cohort belongs to the *Prevotella* genus and is a cluster of nine short-read MAG from our own samples. It exists in 0.10% of the internal validation samples with a mean relative abundance among those who had it of 11.68%, and a maximal relative abundance of 23.40%. However, it did not exist in the external validation at all. The most abundant novel species in the external validation cohort belongs to the *Ruminococcus* genus and is a singleton short-read MAG from our own samples. It exists in 0.13% of the external validation samples with a mean abundance among those who had it of 4.93%, and a maximal relative abundance of 7.30%. However, it did not exist in the internal validation at all. These discrepancies exemplify the difference in microbial composition between different populations and how our reference set, produced from a uniquely large cohort, was even able to find previously unknown species that are very rare in our own cohort's population. The second most abundant novel species is shared by both validation cohorts (exists in 6.46% of internal validation and 4.30% of external validation), and is also of the *Ruminococcus* genus. A third novel species of this genus has 14 genomes in its cluster, making it the biggest cluster among the novel species. All of the genomes in this species cluster are short-read MAG from our own samples.

The genes in the genomes of these 310 species are significantly enriched for 9 out of the 20 COG functional categories tested ($q < 0.0001$, hypergeometric test) ("Methods")[33]. However, since these genomes are also significantly less annotated (median amount of annotated genes out of all the genes in a genome, 48.59% in the novel species vs. 52.80% in the known ones, $P < 0.0001$, Mann–Whitney $U$ test), as expected for newly discovered microbes, this is a biased comparison.

Of the annotated genes in the new species, 42 code for gene products that are not present in the known species of our reference set, some of these products may directly affect the host while others are more beneficial for microbial competition. For example, among those that can directly affect the host, we found Lp49 antigen which is a membrane-associated protein recognized by antibodies present in leptospirosis patients, and Fragilysin, a

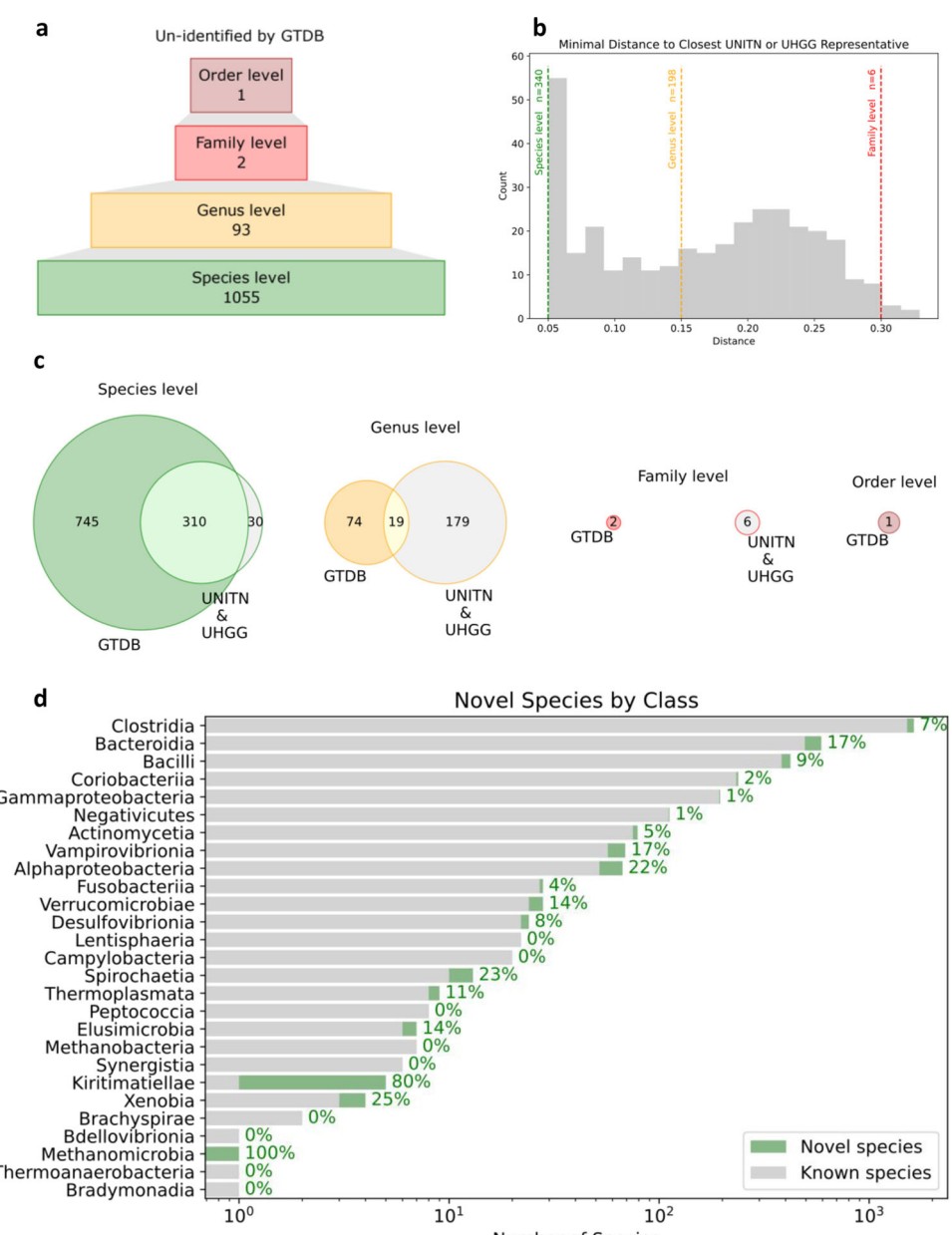

**Fig. 6 310 newly discovered human gut microbial species.** The novel species, stratified by their phylogenetic disparity from previously discovered species —species level in green, genus in orange, family in red and order in brown. **a** Upside-down funnel diagram of genomes un-identified by the Genome Taxonomy Database (GTDB) at some phylogenetic level. **b** Histogram of the WIS representatives' MinHash (MASH) distance from the closest UNITN or UHGG representatives that exceed the species-level distance. Dashed lines are phylogenetic distance thresholds. *n* is the number of genomes that passed each threshold. **c** Venn diagrams of novel species as un-identified by GTDB and relative to the UNITN and UHGG reference sets. Circles scaled to the amount of species in them. **d** Number of known and novel species in each class. The percentage to the right of the bar is the proportion of novel species out of all the species of the class (within our reference set).

metalloprotease toxin that is cytopathic to intestinal epithelial cells and induces fluid secretion and tissue damage in ligated intestinal loops. An example of a gene product that may affect microbial competition is ParE3 toxin, involved in plasmid partition[34,35].

Furthermore, the novel species hold significantly fewer potential antibiotic resistances than the known species (21.94% of the novel species vs. 30.21% of the known ones hold at least one potential antibiotic resistance, and on average there are 1.06 resistances per genome in the novel species and 2.18 in the known ones, $P < 0.01$, Mann–Whitney *U* test) ("Methods" and Supplementary Data 5), yet seven novel species hold more than ten potential resistances out of the 88 tested.

The subset of previously unknown species we elaborated on and their new gene products and antibiotic resistances mentioned are just a few examples of how they may be used to study the connection between the gut microbiome and its human host, and the competition between the microbes that harbor our body.

## Discussion

In this work, we created a human gut microbiome genome reference set (the WIS reference set) that includes 3594 species genomes that belong to 2365 genera and 628 families. We found that 310 of our genomes belong to microbial species that were not known before, 19 of which belong to previously unfamiliar genera.

Our process is based on the previously published work in ref. [7] but differs from it in several ways[7]. First, our genomes were assembled from a uniquely large human gut microbiome set of 51,052 samples, that we further complemented with genomes from multiple countries to create a broad starting assembly set. Second, we introduced genomes from alternative methods based on nanopore sequencing and isolates, to produce genomes that exhibit higher-quality measures. Finally, we applied harsher quality control criteria, including some to overcome known biases, and focused purely on the human gut.

We showed that our reference set recapitulates significantly more reads from stool samples of western populations than the UNITN reference set, and even more than the new UHGG set. In addition, our reference set has a significantly higher number of identified coding regions per genome than the other two sets.

By performing metagenomic assembly on a large cohort, we were able to obtain assemblies for genomes which only rarely appear in a high enough abundance for the assembly process. By the in depth methods, nanopore and isolates, on a small set of samples, we were able to obtain genomes which typically appear only in low abundance, but are common to many individuals. The combination of the two sources allowed a unique coverage of previously unknown species.

Many of these previously unknown species are singletons in our data set, having been assembled from a single sample, although other samples do contain them. Applying advanced culturing and sequencing techniques to such samples may allow acquiring more assemblies of these species and to better the understanding of their function in the human microbiome.

These species add new members to almost all phyla known to exist in the human gut microbiome, and for some phyla, the new species represent over 20% of the genomes they have in our reference set. As an example, in the Kiritimatiellae class, we found an unknown genus with four new species. The most prevalent of our novel species is present in over 15% of samples in both an internal and an external validation cohorts. In contrast, the most abundant novel species differs between these two cohorts. Overall, the 310 novel species encode over 40 gene products that are not seen in any of our known species genomes.

Our gut microbiome reference set builds upon the work of many others, and we believe it provides a significant step forward. As the field progresses, the cost of more accurate sequencing methods will decrease, and it will be easier to create higher-quality genomes. Furthermore, focusing on samples identified in advance as having the potential to contain unknown species, for example in rural populations or in sick individuals, or ones that have been found to date only in low quality, will allow to step even further. More immediate steps can be to understand the gene contents and function of the species in our reference set, and their possible interactions with the host. Another direction could be to complement the reference set with other types of microbes, such as viruses and fungi. Finally, while the focus on a single environmental niche has numerous advantages in creating a consistent reference set, it is important to create such reference sets for other environments, and compare what they share, and how they diverge, along the path to understanding the symbiosis between host and microbiome.

## Methods

**Ethics statement**. The study was approved by the Institutional Review Board (IRB) ethics committee of the Weizmann Institute of Science (Reference number: 1719-1).

**Data sources**. Multiple sources of assemblies were used in the construction of the initial assembly pool. We insisted on using only assemblies for which metadata was available, with emphasis on the sample being confirmed as a human gut microbiome sample, and on being able to identify multiple samples of a single individual (for reasons which will be elaborated below).

We applied the Pasolli et al. assembly pipeline on our 51,052 stool samples, which had undergone single-end short-read shotgun sequencing. Of these, 70 samples also underwent nanopore sequencing, combined with deep paired-end sequencing, using a dedicated pipeline.

External MAG were taken from:

1. Pasolli et al.—119,449 assemblies with relevant metadata. Samples originating from our own cohort were excluded, as we independently applied the above-mentioned assembly pipeline to them[7].
2. Nayfach et al.—10,116 assemblies which had metadata, and were not in overlap with refs. [7,15].
3. Almeida et al.—no assemblies were used since all assemblies that were not in overlap with ref. [7] lacked necessary metadata[7,36].

External isolate assemblies were taken from:

1. Forster et al. and Browne et al.— 737 and 137 isolates from 20 individuals (these two sources share the same cohort)[18,19].
2. Zou et al.—1520 isolates from 150 individuals[17].
3. Poyet et al.—3632 isolates from 90 individuals[16].

**Nanopore samples**. The 70 samples chosen for Nanopore sequencing were chosen for technical reasons, by the amount and quality of fecal sample available, and not by any criteria which take into account their metagenomic contents.

**MAG assembly procedure**. On the 51,052 single-end human genome filtered sequencing files, gathered in previous studies, we applied an assembly procedure designed in ref. [7]:

1. SPAdes (version 3.10.1) with argument '–only-error-correction' was run for preliminary error correction[37].
2. Megahit (version 1.1.1) with argument '–min-contig-len 1000' was run to build contigs[38].
3. Bowtie2 (version 2.2.9) was run to build an index from the contigs file with default arguments, and to map the original sample to this index, with '–very-sensitive-local' argument[39].
4. SAMTools (version 1.3.1) was used to create a sorted bam from the bowtie output[40].
5. Metabat2 (version 2.12.1) with argument '-m 1500' was run, with a depth file created using jgi_summarize_bam_contig_depths with default arguments, to bin contigs into assemblies[41].
6. Quality of initial assemblies was assessed with:

    a. CheckM lineage_wf (version 1.0.13) with default arguments was used to determine quality parameters. With only assemblies with >70% completeness and less <5% contamination taken[21].
    b. Prokka (version 1.12) with default arguments was used to create gff files for each assembly which passes initial quality thresholds[42].
    c. CMSeq (version 1.2) polymut with arguments '–mincov 10 –minqual 30 –dominant_frq_thrsh 0.8' was used to assess heterogeneity. With only assemblies with less than 5% heterogeneity taken[7].

**Quality control**. We follow Passoli et al. in using a more stringent <5% contamination criteria, than the widely used MIMAG standard parameters for medium (contamination <10%, completeness ≥50%) genome quality[7,43]. However, we increased the completeness threshold to >70%, since according to simulations we performed (creating partial assemblies from full single-contig assemblies) at 50% completeness the MASH distance between two assemblies may still be over 0.05, even with no contamination or single position errors. When we apply >70% completeness this no longer happens, and two partial assemblies of the same species do cluster together.

**MASH distances**. We follow Passoli et al. in using MASH as a distance metric[7,24]. MASH is a general-purpose toolkit that utilizes the MinHash technique to estimate genomic distance. MASH distance is a good proxy for one minus the average nucleotide identity (ANI), so that the MASH species-level threshold of 0.05 is equivalent to the widely accepted 95% ANI, used to form species boundaries.

**Focus on the human gut microbiome**. We decided to focus on a single ecological niche, the human gut microbiome. The need for such a focus stems from the process used to define phylogeny, and more specifically for the hierarchical clustering upon which thresholds for species, genus and family levels are defined. Different ecological niches may host the same species, but the set of genes in the genome of the microbe may vary, causing the distance between different assemblies

of the same species to be higher than if a single niche was studied. This may cause single species to be split between different clusters.

**Biases reduction**. Another source of bias, that might stem from the use of hierarchical clustering, especially when the distance between groups is calculated using the average distance between the members of the groups, stems from duplication. Duplicated, or almost duplicated, assemblies will cluster early on (because the distance between the pair is 0 or very close to it), but from there on, this point in space will be given double the normal weight in any averaging step.

In a single-assembly process, from a single sample, such a problem does not come about, as the assembly process and binning of contigs, would not allow two very close assemblies to be created. But, if the process of assembly is performed twice, on the same sample, the two sets of assemblies created will not be exact duplicates, but would be very close to one another, and thus using both sets in the clustering process will cause such a bias. This is also true if two different samples of the same individual are used. The longer the time between samples, the more the microbes of an individual may have evolved; some microbes will evolve only slightly, others may be lost or gained, and the abundances may change so as to make different microbes abundant enough to be assembled at different time points. Assembling microbes from a few samples of the same individual may add new microbes to the set assembled from a single sample, but will also inevitably create a few copies of the same species, which will therefore bias the clustering process.

In order to acquire as many assemblies as possible, without causing such a bias in clustering, we decided to process multiple samples of the same individual, but only take a single assembly of each species. All assemblies of a single individual (that pass quality thresholds) were ranked by quality (see below), and from highest to lowest each was added to the pool of assemblies only if it was above the species-level distance threshold (see above) from all previously added assemblies. For applying this criterion, we needed metadata which allows identifying all the samples taken from the same individual.

This non-biasing criterion was also used for the choice of the isolates' assemblies, where many plates may be growing the same microbe, if they originated from the same stool sample of a single individual.

**Chosen assemblies**. Applying the criteria above reduced the number of assemblies we had at our disposal to:

- 142,912 short-read MAG assembled in-house.
- 86,191 short-read MAG from ref. [7].
- 7211 short-read MAG from ref. [15].
- 2855 nanopore MAG.
- 1327 isolate assemblies from ref. [17].
- 528 isolate assemblies from refs. [18,19].
- 94 isolate assemblies from ref. [16].

**Cluster representatives**. The set of assemblies in a cluster may range from a single assembly to almost ten thousand assemblies. We chose a cluster representative genome by quality, which is not a single criteria, but a balance between different characteristics of the assembled genome. We estimated the completeness, contamination, and N50, and balanced their relative contributions, using the following formula:[21]

$$\text{Completeness} - 5*\text{Contamination} + 15\log_{10}(\text{N50})$$

**Genome identification**. We did not use named "NCBI-RefSeq" sequences in our clustering process as Passoli et al., thus we had to identify our genomes in a different way[7]. We chose to use GTDB-tk version 1.7.0, database version R202 on the species representative genomes for this task.[32,44]

Furthermore, we measured the distances between the representative genomes in the WIS reference set and the genomes of the UNITN and UHGG reference sets, using MASH with $10^4$ sketch size, and their placement in UNITN non-human clusters using PhyloPhlan, phylophlan_metagenomic version 3.0.35 with arguments -n 1 -d SGB.Jan19[45].

**Genetic annotations**. The representative genomes were annotated using prokka version 1.14.6 with the argument –rfam and using eggnog version 2.0.4 with the arguments –go_evidence all and -m diamond[42,46].

**Assessing chimeras**. We used Genome UNClutterer (GUNC) version 1.0.5 with default arguments to assess the probability of a genome being a chimeric mixture of distinct lineages[23].

**Antibiotic resistance**. We used the ABRicate tool[42] version 1.0.1 on the representative genomes with the arguments –db NCBI, CARD, ARGANNOT, RESFINDER (all pulled on June 6th, 2021) –minid 70 –mincov 70 and combined the results of the multiple databases in cases where the sequence was from the same

contig, strand and within a start or end buffer of 100 base pairs. In addition, we united different codes of the same drug[47–50].

**Alignment**. All validation samples are whole-genome sequencing of fecal metagenomic samples which underwent a pipeline of quality assurance, trimming to 100 bps and filtering of human reads. Non-western samples were paired-end but were treated as single-end (i.e., each side separately, as an independent read). Read alignment of metagenomic samples to representative sets was performed using bowtie2 version 2.3.4.2, with arguments -a –no-unal –no-sq –no-hd –score-min L,−40,0. Unique read alignment was defined by reads that are best aligned to a single mapping position[39].

**Relative abundance**. Relative abundance and prevalence estimations were calculated using the unique relative abundance (URA) technique[14]. In broad terms, this method uses only uniquely mapped reads in order to assess abundances. It divides the reference genome into windows of the same number of expected unique positions (positions from which a read taken would be unique in the given reference set), and assesses the distribution of the number of mapped reads on these genome windows. A species exists in the sample if enough reads are uniquely aligned to it, and the distribution of these reads on the genome windows is uniform enough.

**Statistics**. All statistical analyses were conducted using a two-sided Mann–Whitney $U$ test, and Bonferroni correction for multiple hypotheses, unless otherwise stated. 0.05 alpha threshold was used on corrected $P$ values. No statistical method was used to predetermine sample size, we took all available data (up to 2019) after quality control and bias reduction as described above.

**Instruction for using our code to build a reference from a set of assemblies**. Creating a new reference set, is done in several steps, code for which is provided with this paper (GutReferenceSet/Build_Species_Set, see Code availability):

1. Create a list of assembly files and their metadata. This file should be named full_metadata.csv with index which is the full path of the assembled genome fasta file and the following columns:

   a. Source—source of data, so that Source + RegistrationCode is a unique identifier of an individual from which assemblies were created
   b. Method—assembly creation method (MAG/isolate/nanopore...)
   c. AssemblyName—a unique identifier of the assembly
   d. SampleName—an identifier of the sample the assembly was created from (so as to identify assemblies originating from the same sample)
   e. RegistrationCode—an identifier of the individual the sample was taken from (so as to identify assemblies originating from the same individual)
   f. DoNotTake—a column which is either empty or includes reasons not to consider the assembly

2. Calculate the quality of the genomes by running checkM. The code for this stage is GutReferenceSet/Build_Species_Set/quality.py
3. Filter by the quality and by criteria of not taking the same species from the same person twice. The code for this stage is GutReferenceSet/Build_Species_Set/filter.py
4. Mash all vs. all into a memory map. The code for this stage is GutReferenceSet/Build_Our_Set/distance.py
5. Cluster based on memory-mapped distances. The code for this stage is GutReferenceSet/Build_Species_Set/hierarchical_clustering.py
6. Choose a representative genome for each cluster. The code for this stage is GutReferenceSet/Build_Species_Set/choose_representatives.py
7. Name the representative genomes using GTDB. The code for this stage is GutReferenceSet/Build_Species_Set/naming.py
8. Compare the representative genomes to UNITN and build a tree structure. The code for this stage is GutReferenceSet/Utils/phyphlan.py

**Reporting summary**. Further information on research design is available in the Nature Research Reporting Summary linked to this article.

## Data availability
The 142,912 assembled genome fastas used in this study are available in the Zenodo database, divided into ten tarred and gzipped files, in addition to the full metadata file describing these assemblies: https://doi.org/10.5281/zenodo.5767857. The IRB committee determined that bona fide researchers wishing to gain access to the data should be listed and thus access will be provided upon request. The 3594 species representative genomes of this study have been deposited in the Figshare database, in addition to all supplementary files: https://doi.org/10.6084/m9.figshare.16885261.

## Code availability
The source code used to create our reference set is available in the following git repository: https://github.com/erans99/GutReferenceSet.

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

## Acknowledgements

We thank LifeLines for providing a valuable external validation, Yuval Bussi for providing the nanopore assemblies, Yuval Bussi and Amit Lavon for guidance in setting up the genetic annotations and antibiotic resistances processes, and to Adina Weinberger and Itzhak Pilpel for many fruitful discussions along the way. E.S. is supported by the Crown Human Genome Center; Larson Charitable Foundation New Scientist Fund; Else Kroener Fresenius Foundation; White Rose International Foundation; Ben B. and Joyce E. Eisenberg Foundation; Nissenbaum Family; Marcos Pinheiro de Andrade and Vanessa Buchheim; Lady Michelle Michels; Aliza Moussaieff; and grants funded by the Minerva foundation with funding from the Federal German Ministry for Education and Research and by the European Research Council (786344) and the Israel Science Foundation (2970/20).

## Author contributions

S.L. and S.S. conceived and designed the analysis, contributed analysis tools, performed the analysis, and wrote the paper. D.R. conceived and designed the analysis and M.G. contributed analysis tools. E.S. conceived, designed and supervised the analysis, contributed the data, and wrote the paper.

## Competing interests

S.L., S.S., and D.R. declare no competing interests. M.G. reported being an employee of DayTwo. E.S. reported being a paid consultant of DayTwo.
