## [Peer Review File · Nature Communications]

REVIEWER COMMENTS

Reviewer #1 (Remarks to the Author):

The manuscript by Leviatan et al. presents a novel and expanded catalogue of 3,594 species clusters from more than 240 thousand MAGs reconstructed from more than 50 thousand metagenomes.

The main focus of this work is the comparison of the newly generated gut microbiome species clusters against two other published catalogues: UINTN and UHGG. Although many of the comparisons are reported for the WIS vs. the UNITN catalogues, it should be pointed out more clearly that UHGG actually comprises the UNITN catalogue. This will also make several comparisons not significant anymore, for instance, Fig. 2B/D/E.

One of the main points is that this new catalogue is based on 6 times more metagenomes than the UNITN one (51,052 vs. 9,428), although it should be noted that the authors retrieved about 4 times fewer MAGs per metagenome ($483,192 / 51,052 = 9.46$ vs. $345,654 / 9,428 = 36.66$, $36.66 / 9.46 = 3.88$). I think these extensive metagenomic assembly applications are really important, as their application on more and more data will allow uncovering very low-abundant and low-prevalent commensal species. As also, in this case, the authors were able to observe 340 previously undescribed genomes, which would deserve further investigations, like how prevalent they are in general in the human gut microbiome.

Fig. 1, the phylogenetic tree looks a bit weird. It seems that none of the 483,193 MAGs is assigned to Archaea, which could also help visualization by setting the root of the phylogeny in between Archaea and Bacteria. Could this be a result of other chosen cut-offs? I think it will be worth discussing this point.

Fig. 2A, I feel like the comparison on completeness is biased with the respect to the different cut-offs, as also briefly stated in the text, so I think the p-value might be biased and misleading, considering that the WIS catalogue cut completeness >70% and UNITN at >50%. It is true that the authors are comparing the two catalogues, but considering such a difference I'm not sure the p-value is really meaningful in this case.

Fig. 2B/C/E, I feel like the violin plot visualization is not able to cut at 0 and show a non-existing curve below zero. Please, double-check, as values below 0 would be very strange in these cases.

Lines 266-268. Although it is true that the UNITN set comprises more body sites, it should also be noted that the majority of samples are from the gut and only a small fraction of them are from other body sites.

Reviewer #2 (Remarks to the Author):

Review done by: Se Jin Song, UC San Diego

Summary

Leviatan et al. seeks to improve on the human microbiome reference of MAGs generated by Pasolli et al. 2019 by adding data from additional samples, including nanopore and isolate data, and focusing on the gut microbiome. They perform a benchmark of this expanded reference against the original Pasolli reference and a second reference called UHGG.

Overall, I think the manuscript shows that this improvement is achieved, and I commend the authors for the work. However, it was not evident to me--until reading the methods--that the WIS reference set includes the assemblies from the Pasolli et al (minus those from non-gut samples). I believe the language throughout can be adjusted to more clearly reflect that this is an improvement of an existing reference, as opposed to being an entirely different one. It also wasn't clear whether some of the same assemblies are in the UHGG reference set. While Table 1 is helpful, it would be useful to show how much overlap there is among the reference sets in terms of underlying data. Moreover, the first time we see UHGG is in Table 1, without context as to why it has been included as a comparison. Introducing the UHGG dataset much earlier would be useful, perhaps explaining in the introduction that the work here is a benchmark of an improved reference against the reduced reference and a second distinct reference, using validation datasets representing both western and non-western populations.

What is also lacking is an explanation or demonstration of how better mapping might lead to insights that were not possible before. For example, the validation datasets could have been used to demonstrate how some result or interpretation of the result may change based on the reference used. Further regarding the validation datasets, I suggest adding the content from Figure S2 into Figure 3. I would have also appreciated a greater discussion about why these datasets may have performed the way they did and ways to improve mapping rates more consistently across datasets representing a wide range of populations.

The manuscript could also benefit from more focus in both the introduction and discussion about how reference sets may be continuously improved by the addition of data from more samples, different data types (longer read technology, data from isolates), and the importance of stricter curation and quality control measures or at least reference other publications that may discuss these issues in further detail.

More specific comments are provided below.

A q-value is reported in many places. Was this intended to be p-value?

Lines 44-47:

I wouldn't say that 16S is very limited by known sequences.

And even for shotgun data sequencing, MAGs are now more routinely generated. However, assigning taxonomy to either is limited by reference dataset. I recommend clarifying these points. I also think a thought bridge between this statement and the next paragraph is needed, stating how and why both of these methods can benefit greatly from more comprehensive and quality references.

Line 139:

Change 'less' to 'fewer'

There are many more instances of this. Please check usage throughout.

Lines 183-186

Is another possibility that some of MAGs represented by the Tanzanian samples may have been removed by the quality cutoffs?

266-273

Another reason could be that annotated databases are likely largely biased towards functions in the human gut microbiome.

"Regardless of the reason, this indicates that for the gut microbiome our new reference set covers a wider range of genes and hence functions than the UNITN reference set."

I don't see how this is supported. In sheer number, UNITN still has a larger number of uniquely annotated genes. Please clarify.

Line 291

I'm unsure of what is meant by 'it requires prior knowledge to tie the variable 16S regions' sequences to a particular species'.

It can often be placed within a group (although not to the species level), but that has to do with resolution, not the reference per se.

Methods

Line 583: typo? Why note the dataset if 0 assemblies were used from Almeida et al? Or is it to point out that these all overlapped with Pasolli et al?

We thank the reviewers for the useful comments. We have revised our manuscript and addressed the comments raised by the reviewers and editor, and we believe that the manuscript is now substantially improved. Below is a point-by-point response to reviewers' comments, and a new manuscript, with changes highlighted, is attached. In addition to the points mentioned we added a new "Instruction for using our code to build a reference from a set of assemblies" subsection to the methods section.

We believe the revised manuscript is in line with the comments and suggestions raised by the reviewers. We hope that you will find our revised manuscript suitable for publication.

Reviewer #1 (Remarks to the Author):

The manuscript by Leviatan et al. presents a novel and expanded catalogue of 3,594 species clusters from more than 240 thousand MAGs reconstructed from more than 50 thousand metagenomes.

The main focus of this work is the comparison of the newly generated gut microbiome species clusters against two other published catalogues: UINTN and UHGG. Although many of the comparisons are reported for the WIS vs. the UNITN catalogues, it should be pointed out more clearly that UHGG actually comprises the UNITN catalogue. This will also make several comparisons not significant anymore, for instance, Fig. 2B/D/E.

Thank you for the comment. We now introduce UHGG both in the introduction and the beginning of the results. We further elaborated that UHGG builds upon the gut segment of UNITN and specified that 48% of its assemblies originated in the UNITN set in Table 1.

Introduction:

"We show our reference set is better for the gut microbiome than the previous benchmark in the field, using validation cohorts representing both western and non-western populations. We also show our reference set challenges a second newly proposed set."

Results:

Section - Building a new reference set

"Our process of creating the human gut microbial genomes reference set follows the work of Passoli et al. (Methods). Their work used 9,428 metagenomic samples from multiple human body sites, although mostly from the gut, to recapitulate 4,930 microbial species, many of which were not known before (Table 1, Figure 1). From here after we term their set of genomes the University of Trento or "UNITN" reference set. We will later refer to a second set, the Unified Human Gastrointestinal Genome collection, as the "UHGG" reference set."

Section - Comparison to UHGG

"A new reference set, expanding the gut segment of UNITN by Passoli et al., was recently published as the Unified Human Gastrointestinal Genome collection."

One of the main points is that this new catalogue is based on 6 times more metagenomes than the UNITN one (51,052 vs. 9,428), although it should be noted that the authors retrieved about 4 times fewer MAGs per metagenome ($483,192 / 51,052 = 9.46$ vs. $345,654 / 9,428 = 36.66$, $36.66 / 9.46 = 3.88$). I think these extensive metagenomic assembly applications are really important, as their application on more and more data will allow uncovering very low-abundant and low-prevalent commensal species.

We acknowledge this requires elaboration. There are several reasons for the lower ratio of assemblies per sample as we now explain it in this new paragraph of the manuscript.

Results:

Section - Comparison to UNITN

“We procured fewer assemblies per sample than UNITN for several reasons. Our samples are all single-end with $9.74e+08 \pm 3.18e+08$ bp per sample (computed by read length times read depth), while UNITN comprises both single-end and paired-end samples with $4.21e+09 \pm 3.39e+09$ bp per sample, this lower number of bp restrict the number of assemblies that can be produced from a sample with current computational tools. Furthermore, applying harsher quality control and filtration criteria reduced the number of assemblies we obtained from each sample.”

As also, in this case, the authors were able to observe 340 previously undescribed genomes, which would deserve further investigations, like how prevalent they are in general in the human gut microbiome.

We agree this is an interesting point. Data regarding all species composition in the validation cohorts is provided as supplementary table 5. Details regarding the most abundant and most prevalent novel species are included in the manuscript under the results section. Additionally, below are figures that allow exploration of the novel and previously known species prevalence and relative abundance that we based the text upon.

Results:

Section - Novel human gut microbial genomes

*“The most prevalent novel species in both the internal (15.02% of the samples had it with a mean relative abundance among those who had it of 0.10%, and a maximal relative abundance of 1.02%) and external (18.98% of the samples had it with a mean relative abundance among those who had it of 0.11%, and a maximal relative abundance of 0.85%) validation cohorts is a species of the genus *Faecalibacterium*. It is a singleton short-read MAG from Nayfach et al.¹⁵ Another novel species of this genus is third in its prevalence in the internal validation, and second in the external validation cohort.*

*Among those who had it, the most abundant novel species in the internal validation cohort belongs to the *Prevotella* genus and is a cluster of nine short-read MAG from our own samples. It exists in 0.10% of the internal validation samples with a mean relative abundance among those who had it of 11.68%, and a maximal relative abundance of 23.40%. However, it did not exist in the external validation at all. The most abundant novel species in the external validation cohort belongs to the *Ruminococcus* genus and is a singleton short-read MAG from our own*

samples. It exists in 0.13% of the external validation samples with a mean abundance among those who had it of 4.93%, and a maximal relative abundance of 7.30%. However, it did not exist in the internal validation at all. These discrepancies exemplify the difference in microbial composition between different populations and how our reference set, produced from a uniquely large cohort, was able to find novel species that are very rare in our own cohort's population. The second most abundant novel species is shared by both validation cohorts (exists in 6.46% of internal validation and 4.30% of external validation), and is also of the *Ruminococcus* genus. A third novel species of this genus has 14 genomes in its cluster, making it the biggest cluster among the novel species. All of the genomes in this species cluster are short-read MAG from our own samples."

Comments regarding the figures:

The color indicates the phylogenetic disparity

assemblies in the legend are in log10 scale

The abundance is among those who had the species

Numbers adjacent to dots are representative species ID of the 10 highest species in terms of prevalence and the 10 highest species in terms of relative abundance

Fig. 1, the phylogenetic tree looks a bit weird. It seems that none of the 483,193 MAGs is assigned to Archaea, which could also help visualization by setting the root of the phylogeny in between Archaea and Bacteria. Could this be a result of other chosen cut-offs? I think it will be worth discussing this point.

We appreciate this comment which helped us improve the visualization of the phylogenetic tree. Archaea do exist in our assembled genomes, even though not as frequently as Bacteria, to make this point clear we added it to the introduction. In our reference set there are 17 species of the Archaea domain and they span three phyla - Halobacteriota (one species), Methanobacteriota (seven species) and Thermoplasmata (nine species). Two of these species are novel, species ID 7 and 900 and are first revealed by our work, we acknowledge one of them in the results.

The archeal domain has similar representation in the UNITN reference set (18 species) so we do not think the different cutoffs affected their representation in our set. We think this is a consequence of Archaea being a less abundant domain in the gut microbiome than Bacteria.

The phylogenetic tree is of the species representative genomes chosen for each cluster and includes the archaea species. Their phyla in the legend is underlined in red in the figure below and their specific location in the tree is circled in red. The phylogenetic tree was originally

optimally ordered for visualization, we understand how it can be confusing and changed it to be ordered strictly by phylogeny. We also increased its size for better visualization. Unfortunately, we were not able to set the root of the tree to our wish, as the program that draws it does not have such an option.

Introduction:

“In this work, we build upon previously published genomes and our own set of assemblies from 51,052 metagenomic samples, and apply strict quality control and anti-biasing considerations to form a set of 241,118 genomes. We proceed to cluster this set, based on the genomic distances between pairs of assemblies, and choose a representative genome for each cluster. In this way we construct a genome reference set with median completeness of 95% and median contamination of 0.67%, that represents 3,594 gut microbial species. All but 17 of these species are from the Bacteria domain, with the rest being Archaea.”

Results:

Section - Novel human gut microbial genomes

“Three phyla have a single genome in our reference set and of them, one was found to be a novel species of the Halobacteriota phylum, Methanomicrobia class (Figure S3, Figure 6d). This species is of the archeal domain and Methanocorpusculum genus, and it originated from a singleton short-read MAG of our own samples.”

Fig. 2A, I feel link the comparison on completeness is biased with the respect to the different cut-offs, as also briefly stated in the text, so I think the p-value might be biased and misleading, considering that the WIS catalogue cut completeness >70% and UNITN at >50%. It is true that the authors are comparing the two catalogues, but considering such a difference I'm not sure the p-value is really meaningful in this case.

Thank you for the comment. We agree the p-value in this comparison is affected by the cut-offs and thus we address it in the text, to make it as clear in the figures we added a comment addressing the different thresholds used.

Results:

Section - Comparison to UNITN

The notable difference in completeness is not incidental as we set this parameter threshold to be 70% for all of our assemblies while Passoli et al. chose a threshold of 50%.⁷ We chose to increase the completeness threshold, as higher completeness results in lower systematic bias in quality parameters estimates, and this bias is very small (<2%) in cases where genomes are over 70% complete and less than 5% contaminated. Higher completeness also ensures clustering together of partial assemblies of the same genome (Methods)."

Figure 2 and S1

"Completeness threshold in WIS was 70% while it was 50% in UNITN and UHGG."

Fig. 2B/C/E, I feel like the violin plot visualization is not able to cut at 0 and show a non-existing curve below zero. Please, double-check, as values below 0 would be very strange in these cases.

Thank you for highlighting this point. We double checked and as can be seen in supplementary file 1, none of the values are below zero, this is a graphical illusion of violin plots since the distribution is dense around the minimal value (zero). We added a comment for each relevant figure to make it clear.

Figures 2, 4, S1 and S2:

"Panels b, c and e seemingly have values under zero, this is a graphical illusion of violin plots since the distribution is dense around the minimal value (zero)."

Lines 266-268. Although it is true that the UNITN set comprises more body sites, it should also be noted that the majority of samples are from the gut and only a small fraction of them are from other body sites.

Thank you for raising this point. We added clarifications regarding this point to the first introduction of UNITN and to the relevant results section. This is also quantitatively mentioned in Table1 "Gut (85%), oral (8.5%), skin (5.4%), vagina (1%) and maternal milk (0.1%)". Even though UNITN is built mostly from gut samples, its performance for other body sites, such as the oral environment, is much better than the performance of our reference set. This shows that even the relatively small percentage of non-gut samples have successfully covered functionality that does not exist in the gut.

Results:

Section - Building a new reference set

“Our process of creating the human gut microbial genomes reference set follows the work of Passoli et al. (Methods). Their work used 9,428 metagenomic samples from multiple human body sites, although mostly from the gut, to recapitulate 4,930 microbial species, many of which were not known before (Table 1, Figure 1).”

Section - Genetic annotations

“These results are surprising since the UNITN reference set includes genomes that originated from multiple body sites (although most are from the gut) and thus are expected to cover a wider range of functions while the WIS reference set originated only in a single body site - the gut.”

Reviewer #2 (Remarks to the Author):

Review done by: Se Jin Song, UC San Diego

Summary

Leviatan et al. seeks to improve on the human microbiome reference of MAGs generated by Pasolli et al. 2019 by adding data from additional samples, including nanopore and isolate data, and focusing on the gut microbiome. They perform a benchmark of this expanded reference against the original Pasolli reference and a second reference called UHGG.

Overall, I think the manuscript shows that this improvement is achieved, and I commend the authors for the work. However, it was not evident to me--until reading the methods--that the WIS reference set includes the assemblies from the Pasolli et al (minus those from non-gut samples). I believe the language throughout can be adjusted to more clearly reflect that this is an improvement of an existing reference, as opposed to being an entirely different one. It also wasn't clear whether some of the same assemblies are in the UHGG reference set. While Table 1 is helpful, it would be useful to show how much overlap there is among the reference sets in terms of underlying data.

Thank you for raising this point that may help clarify this issue to other readers as well. About one third of the assemblies we use are from Passoli et al., which constitutes most of the external assemblies we added alongside our own. As for UHGG, the use of Passoli et al. is a major part (almost half) of the assemblies. We address this in the text, and added the information in Table 1.

Added to Table 1:

"External assemblies after filtration criteria#

WIS 98,206 (88% Passoli et al.)

UHGG 286,997 (48% Passoli et al.)

Total assemblies used

WIS 241,118 (36% Passoli et al.)

UHGG 286,997 (48% Passoli et al.)"

Results:

Section - Building a new reference set

"Our metagenomic samples are mostly of Israeli adults (range 4-93 years, median 55) but we also complimented the 142,912 newly assembled genomes produced from our samples with assemblies we gathered and curated from previously published studies (up to 2019) originating in different countries (short-read metagenome sequencing based assemblies: 86,191 from Pasolli et al. and 7,211 from Nayfach et al., isolate based assemblies: 94 from Poyet et al., 1,327 from Zou et al., Forster et al. and Browne et al., and nanopore metagenome sequencing based assemblies: 2,855 from Bussi et al.) (Table 1, Methods)."

Section - Comparison to UHGG

"A new reference set, expanding the gut segment of UNITN by Passoli et al., was recently published as the Unified Human Gastrointestinal Genome collection."

Moreover, the first time we see UHGG is in Table 1, without context as to why it has been included as a comparison. Introducing the UHGG dataset much earlier would be useful, perhaps explaining in the introduction that the work here is a benchmark of an improved reference against the reduced reference and a second distinct reference, using validation datasets representing both western and non-western populations.

Indeed table 1 was the first place the UHGG set came up, and should not have been. We now added a general statement on the subject of the other reference sets to the introduction, and introduced the UHGG set, alongside the UNITN set, in the first paragraph of the results.

Introduction:

“We show our reference set is better for the gut microbiome than the previous benchmark in the field, using validation cohorts representing both western and non-western populations. We also show our reference set challenges a second newly proposed set.”

Results:

Section - Building a new reference set

“Our process of creating the human gut microbial genomes reference set follows the work of Passoli et al. (Methods). Their work used 9,428 metagenomic samples from multiple human body sites, although mostly from the gut, to recapitulate 4,930 microbial species, many of which were not known before (Table 1, Figure 1). From here after we term their set of genomes the University of Trento or “UNITN” reference set. We will later refer to a second set, the Unified Human Gastrointestinal Genome collection, as the “UHGG” reference set.”

What is also lacking is an explanation or demonstration of how better mapping might lead to insights that were not possible before. For example, the validation datasets could have been used to demonstrate how some result or interpretation of the result may change based on the reference used.

We thank the reviewer for pointing out that this may not be clear to our readers. We added to the results an explanation of the utility of increased mapping. In previous work our group showed how the use of the UNITN reference set has significantly improved predictive power of species composition on host traits and phenotypes, such as age, gender and HbA1C% (Rothschild et al.). In Single Nucleotide Variations analyses that are still at early stages, we experienced a big jump in accuracy when moving from the UNITN reference set to our own.

Results:

Section - Comparison to UNITN

“Accounting for more of the genetic content, as exemplified by the increased mappability, aids in studying the microbes. In particular, it helps in understanding their genetics and the effect variations within their genomes has on their capabilities and on host health.”

Further regarding the validation datasets, I suggest adding the content from Figure S2 into Figure 3. I would have also appreciated a greater discussion about why these datasets may have performed the way they did and ways to improve mapping rates more consistently across datasets representing a wide range of populations.

Thank you for the suggestion, we have joined the two figures, and expanded more on this point.

Results:

Section - Comparison to UNITN

“We conducted the same process on 3,096 held out samples from our own cohort where we received similar results, and on three smaller cohorts of non-western populations from India (n=110), El Salvador (n=113) and Tanzania (n=68), that are often less represented in scientific research. On the non-western cohorts the difference in read alignment was not significant, and in one case, of the Tanzanian cohort, it was even significantly higher in UNITN, perhaps as Tanzania is relatively highly represented in the UNITN set, and did not necessarily make it into our set. In terms of unique alignment, our significant advantage was maintained throughout the non-western populations tested, which shows our process was broad enough and did not specifically capitulate the typical Israeli adult gut microbiome (Figure 3b-e).”

Discussion:

“Furthermore, focusing on samples identified in advance as having the potential to contain unknown species, for example in rural populations or in sick individuals, or ones that have been found to date only in low quality, will allow all of us to step even further.”

The manuscript could also benefit from more focus in both the introduction and discussion about how reference sets may be continuously improved by the addition of data from more samples, different data types (longer read technology, data from isolates), and the importance of stricter curation and quality control measures or at least reference other publications that may discuss these issues in further detail.

We modified the sections of the text regarding continuous improvements of reference sets and find all the reviewers suggestions to now be covered. If there are other points that we missed please let us know.

Introduction:

“In addition, a grand effort has taken place in past years to improve upon assembling new microbial genomes from metagenomic sequencing, and to establish techniques of classifying these genomes to create more elaborated reference sets of the human microbiome.”

“Our understanding of the microbes residing within our bodies and their effect on our health is limited by our ability to identify the microbial content. The two main paths to microbial research are either through marker genes, such as the 16S ribosomal RNA genes, or through reference alignment of shotgun read sequencing. In both cases, the broad genetic contents of each microbe is deduced from what is known about its genome. Continual efforts are made to expand this knowledge by establishing better quality and more comprehensive genomes, by using innovative methods and more samples gathered from different populations.”

Results:

Section - Comparison to UNITN

“The notable difference in completeness is not incidental as we set this parameter threshold to be 70% for all of our assemblies while Passoli et al. chose a threshold of 50%.⁷ We chose to increase the completeness threshold, as higher completeness results in lower systematic bias in quality parameters estimates, and this bias is very small (<2%) in cases where genomes are over 70% complete and less than 5% contaminated.^{20,21,22} Higher completeness also ensures clustering together of partial assemblies of the same genome (Methods). The notable difference in contigs length expressed by the N50 parameter is driven mostly by the alternative methods for curating genomes we used, nanopore and isolates as will be later shown (Figure 1 third ring, Figure 4c). Our reference set has fewer genomes (3,594 vs. 4,930), which is expected given the single body site used to create it. For example, species that are exclusive to the skin microbiome should not appear in our gut-derived reference set (Table 1)..

Computational approaches of producing genomes can suffer from mis-binning, association of unrelated contigs resulting in chimeric genomes. In order to mass evaluate the frequency of this potential issue, we used Genome UNClutterer (GUNC) that assesses the probability of a genome being a chimeric mixture of distinct lineages by using a clade separation score (CSS) (Methods).²³ We found that in comparison to UNITN our new reference has significantly fewer genomes suspected to be chimeric (6.82% vs. 11.36% CSS>0.45, median CSS 0.09 vs. 0.12, $q < 0.0001$, Mann-Whitney U test) (Figure 2e).”

Discussion:

“By performing metagenomic assembly on a large cohort we were able to obtain assemblies for genomes which only rarely appear in a high enough abundance for assembly. By the in depth methods, nanopore and isolates, on a small set of samples, we were able to obtain assemblies for genomes which typically appear only at low abundance, but are common to many individuals. The combination of the two sources allowed a unique coverage of previously unknown species.”

“Our new gut microbiome reference set builds upon the work of many others, and we believe it provides a significant step forward. As the field progresses, the cost of more accurate sequencing methods will decrease, and it would be easier to create higher quality genomes. Furthermore, focusing on samples identified in advance as having the potential to contain unknown species, for example in rural populations or in sick individuals, or ones that have been found to date only in low quality, will allow all of us to step even further. More immediate steps can be to understand the gene contents and function of the species in our reference set, and their possible interactions with the host. Another direction could be to complement the reference set with other types of microbes, such as viruses and fungi. Finally, while the focus on a single environmental niche has numerous advantages in creating a consistent reference set, it is important to create such reference sets for other environments, and compare what they share, and how they diverge, along the path to understanding the symbiosis between host and microbiome”

More specific comments are provided below.

A q-value is reported in many places. Was this intended to be p-value?

Thank you for pointing out that this is not clear. The q-value is the Bonferroni corrected p-value of the test, now explained clearly the first time the q-value is used.

Results:

Section - Comparison to UNITN

“In comparison to UNITN our new reference set is composed of genomes that are of significantly higher quality (median completeness 95% vs. 91%, $q < 0.0001$ (q is the Bonferroni corrected p-value) and median contamination 0.67% vs. 0.70%, $q < 0.05$, Mann-Whitney U test)”

Lines 44-47:

I wouldn't say that 16S is very limited by known sequences.

And even for shotgun data sequencing, MAGs are now more routinely generated. However, assigning taxonomy to either is limited by reference dataset. I recommend clarifying these points. I also think a thought bridge between this statement and the next paragraph is needed, stating how and why both of these methods can benefit greatly from more comprehensive and quality references.

Line 291

I'm unsure of what is meant by 'it requires prior knowledge to tie the variable 16S regions' sequences to a particular species'.

It can often be placed within a group (although not to the species level), but that has to do with resolution, not the reference per se.

This is an excellent point, as different 16Ss can be identified even without full knowledge of their taxonomy to the species level. Indeed we had a specific use-case in mind - that of using marker genes (16S and others) in order to deduce the full species genomic contents. We hope the new phrasing makes the point we wanted to make easier to understand.

Introduction:

“Our understanding of the microbes residing within our bodies and their effect on our health is limited by our ability to identify the microbial content. The two main paths to microbial research are either through marker genes, such as the 16S ribosomal RNA genes, or through reference alignment of shotgun read sequencing. In both cases, the broad genetic contents of each microbe is deduced from what is known about its genome. Continual efforts are made to expand this knowledge by establishing better quality and more comprehensive genomes, by using innovative methods and more samples gathered from different populations.”

Results:

Section - Genetic annotations

“The 16S gene is of particular interest as it is known to be present in all bacteria and consists of a highly conserved region that can be identified using PCR, interspersed by variable regions that are species or genus specific. The downside of identifying bacteria using 16S is that it requires prior knowledge to tie the variable 16S regions' sequences to a taxonomy, and that it only recognizes bacteria presence and not its full genomic context. For many of the gut bacteria this prior knowledge does not exist but can now be deduced using our improved quality expanded reference set.”

Line 139:

Change 'less' to 'fewer'

There are many more instances of this. Please check usage throughout.

Changed in 6 relevant locations including the one pointed here.

Lines 183-186

Is another possibility that some of MAGs represented by the Tanzanian samples may have been removed by the quality cutoffs?

Thank you for raising this point. We indeed suspect that removing Tanzanian assemblies which were not of high enough quality contributed to its lower mapping percentage in Tanzanian samples. However, this is not the only possible explanation, as other issues may have driven the differences, such as lack of relevant metadata of some of the Tanzanian assemblies, differences in clustering due to a different set of assemblies being used, or the criteria for choice of representative from each cluster. And most probably, a combination of all these issues. We added this point to the manuscript.

Results:

Section - Comparison to UNITN

"On the non-western cohorts the difference in read alignment was not significant, and in one case, of the Tanzanian cohort, it was even significantly higher in UNITN, perhaps as Tanzania is relatively highly represented in the UNITN set, and did not necessarily make it into our set."

266-273

Another reason could be that annotated databases are likely largely biased towards functions in the human gut microbiome.

"Regardless of the reason, this indicates that for the gut microbiome our new reference set covers a wider range of genes and hence functions than the UNITN reference set."

I don't see how this is supported. In sheer number, UNITN still has a larger number of uniquely annotated genes. Please clarify.

Thank you for raising this possible explanation, we now added it to the results. Although most genomes and thus annotations in UNITN arise from gut samples, its performance for other body sites, such as the oral environment, is much better than the performance of our reference set. This shows that even the relatively small percentage of non-gut samples have successfully covered functionality that does not exist in the gut.

Results:

Section - Genetic annotations

"These results are surprising since the UNITN reference set includes genomes that originated from multiple body sites (although most are from the gut) and thus are expected to cover a wider range of functions while the WIS reference set originated only in a single body site - the gut. A possible explanation for these results is that the increased quality of the WIS genomes made them more accurate and thus it was easier for the annotation tools to recognize the genes in the reference set. Another possible complementary explanation is that the higher number of starting samples allowed the curation of genomes and hence functions of microbes

that are missing from the UNITN reference set. In contrast, this could stem from the gut being a more researched body site, and hence more annotated. Regardless of the reason, since the absolute number of uniquely annotated genes is comparable this indicates that, specifically for the gut microbiome, our new reference set covers a wider range of genes and hence functions than the UNITN reference set.”

Methods

Line 583: typo? Why note the dataset if 0 assemblies were used from Almeida et al? Or is it to point out that these all overlapped with Pasolli et al?

This is not a typo as we now clarify in the text. Almeida et al. is an important possible source of assemblies, and we want to make sure the readers do not think we have missed it. Unfortunately, after careful consideration of all the sources used by Almeida et al. we found that a great deal were in overlap with the sources used by Pasolli et al., and the rest were lacking the required metadata for our anti-biasing consideration. Had we started with Almeida et al. and only then added Pasolli et al., then most of the assemblies would have been attributed to Almeida et al., thus have credited it as a source.

Methods:

Section - Data Sources

“Almeida et al. - no assemblies were used since all assemblies that were not in overlap with Pasolli et al. lacked necessary metadata.”

REVIEWERS' COMMENTS

Reviewer #1 (Remarks to the Author):

This reviewer thanks the authors for the amended manuscript in which all raised points were answered.

Very minor point, the phylogenetic tree in Fig. 1 still looks weird, as usually trees are rooted between the Archaea and Bacteria subtrees, while it seems that the new phylogeny has been randomly rooted. Please consider re-rooting the phylogeny for clarity.

Reviewer #2 (Remarks to the Author):

I thank the authors for the revision, and adding in instructions for using the code.

All of my concerns have been addressed.

At this stage, I recommend minor editing to improve phrasing and to use language that is more objective.

Some examples are provided below.

line 74

recommend 'We sought to improve the previously published set...'

line 174

recommend replacing 'means' with 'suggests'

Figure 2 description

recommend 'The WIS reference set shows higher quality measures than UNITN'

line 234

recommend 'with data from Oxford nanopore sequencing and sequencing of isolates, both methods known to produce higher quality assemblies (REFS)'

line 245

recommend 'significantly lower than the nanopore genomes'

line 404

recommend 'combined into a more comprehensive genome set' or similar

line 430

recommend '19 of which belonged to genera that may be considered novel'

Response to REVIEWERS' COMMENTS

Reviewer #1 (Remarks to the Author):

This reviewer thanks the authors for the amended manuscript in which all raised points were answered.

Very minor point, the phylogenetic tree in Fig. 1 still looks weird, as usually trees are rooted between the Archaea and Bacteria subtrees, while it seems that the new phylogeny has been randomly rooted. Please consider re-rooting the phylogeny for clarity.

Thank you for raising this point, unfortunately we have still not been able to control this attribute. We wrote to the creators of the program that draws the tree, if we get a response before publication we will be more than happy to re-do the tree and update the editor.

Reviewer #2 (Remarks to the Author):

I thank the authors for the revision, and adding in instructions for using the code.

All of my concerns have been addressed.

At this stage, I recommend minor editing to improve phrasing and to use language that is more objective.

Some examples are provided below.

line 74

recommend 'We sought to improve the previously published set...'

line 174

recommend replacing 'means' with 'suggests'

Figure 2 description

recommend 'The WIS reference set shows higher quality measures than UNITN'

line 234

recommend 'with data from Oxford nanopore sequencing and sequencing of isolates, both methods known to produce higher quality assemblies (REFS)'

line 245

recommend 'significantly lower than the nanopore genomes'

line 404

recommend 'combined into a more comprehensive genome set' or similar

line 430

recommend '19 of which belonged to genera that may be considered novel'

Thank you for the improvements in phrasing, we have accepted them all.